# A Simple Method of Evaluating the Thermal Properties of Metallurgical Cokes under High Temperature

**DOI:** 10.3390/ma14195767

**Published:** 2021-10-02

**Authors:** Guangzhi Yang, Xiaoqiang Wang, Ting Shi, Xinci Wu, Yuhua Xue

**Affiliations:** School of Materials Science and Engineering, University of Shanghai for Science and Technology, Shanghai 200093, China; XQ320557@163.com (X.W.); shiting_usst@163.com (T.S.); wxc561@163.com (X.W.)

**Keywords:** coke, tumbling strength, reactivity index, optical texture

## Abstract

The reactivity index of weight loss (RI) and tumbling strength after the reaction (I_10_^600^) of manufacturing coke were first tested at a temperature series of 1100, 1200, and 1300 °C under CO_2_ atmosphere with different compositions and duration times to study the effects of temperature, time, and gas composition on coke hot strength. Then the RI/I_10_^600^, carbon structure, and optical texture of the cokes prepared from different single coals were mainly studied after a solution reaction with CO_2_ under a high temperature of 1300 °C and a standard temperature of 1100 °C. It was found that temperature greatly affects the RI/I_10_^600^ of coke, especially at high temperatures up to 1300 °C. Compared with standard tests under 1100 °C, the changes of RI/I_10_^600^ for different cokes are very different at 1300 °C, and the changes are greatly related to coke optical texture. Under a high temperature in the testing method, the tumbling strength of cokes with more isotropy increased, whereas it decreased for those with less isotropy. This simple method of using high temperature could yield the same results when compared with complicated simulated blast furnace conditions.

## 1. Introduction

Coke is a basic, essential, and irreplaceable raw material for blast furnace (BF) ironmaking, in which it plays four major roles, namely reductant, carbon source, fuel, and frame support. It is very important to use cokes with appropriate reactivity and high strength for the stable running of a BF. The reactivity and strength of coke are often evaluated by reactivity index (CRI) and strength after reaction (CSR) in a standard testing method. CRI is determined by exposing coke to pure carbon dioxide for solution reaction at 1100 °C for 2 h and measuring the coke weight loss afterward. CSR is determined by weighing the ratio of reacted coke with +10 mm sizes after being tumbled for 600 revolutions in an I-drum. CRI/CSR is widely used to evaluate the quality of coke by steel industries around the world. Many factors that affect the CRI/CSR values of coke have been studied, including mineral composition, reflectance parameters, porous structure, and optical texture [1,2,3,4].

However, the reaction condition of standard tests is different from the actual experiences of coke in BFs. At first, the actual reaction temperature in the BF ranges from 800 °C to 1400 °C. Secondly, the reaction gas of CO_2_ is not pure and its composition changes with the descent of coke from top to bottom in the BF. Finally, the reaction time lasts longer than 2 h. Thus, evaluating the accuracy of CRI/CSR for coke quality in BFs has been disputed for a long time, especially in recent years since some researchers have found that the coke quality of CRI/CSR is not in good accordance with BF practices or experimental BF conditions. Many researchers have revised the CRI and CSR test conditions, including changing the pure CO_2_ to a mixture, raising the reaction temperature to a higher temperature, controlling for a constant weight loss of coke, etc. Most found a poor correlation between the results of simulated conditions and standard conditions, or even a mistake in the estimation of coke quality by CRI/CSR. They proposed that the postreaction strength test should be modified in accordance with the individual BF operation, especially at high temperatures [5,6,7,8,9,10,11,12,13,14].

In our former publication [15], we reported our revised simulated conditions for coke evaluation at high temperature. The reaction temperature was up to 1350 °C and the total reaction time was about 334 min. We found that changes in the strength of coke after a reaction are very different for different cokes. Under simulated conditions compared with standard methods of CRI/CSR, some cokes have higher strengths while others have lower ones, leading to a big difference in evaluating coke strength under different conditions.

In the following study, we investigated more samples in our laboratory and found the same results, which further confirms that the standard NSC method does have some limitations in simulating coke degradation behavior in BFs and evaluating the thermal properties of coke.

Since the actual operating conditions of a BF are complicated and it is difficult to identically replicated these conditions in simulation, different scholars have their own understandings about the simulations. The experimental conditions are all complicated compared with the standard NSC method, and even special equipment is required.

In this paper, a simple method of testing coke under high temperature conditions is proposed to replace the complicated simulation methods. We hope to provide a simple method with a better result compared with the standard NSC method. Because high temperature accelerates the rate of coke solution loss reaction and causes the tumbling strength of coke decrease quickly, the proper conditions were first investigated by studying metallurgical coke from blended coals. Then, eight cokes from single coals of different volatiles were tested. The changes in tumbling strength, carbon structure, and optical texture of the cokes were then studied. Afterward, the main differences in the changes were compared, and the possible reasons for the differences were proposed.

## 2. Materials and Methods

### 2.1. Samples

Samples of one metallurgical coke from blended coals and pilot oven cokes from single coals of different volatiles were studied here. All the cokes were provided by Baosteel Company in China. The proximate analyses of the blended and single coal samples for the cokes are shown in Table 1. Among them, C1 is blended coal for metallurgical coke and the others are single coals for pilot oven cokes with different volatiles. C1 was coked in a manufacturing coke oven and those of C2 –C9 were coked in a 70 kg pilot oven.

### 2.2. Solution Reaction Tests and Characterization

Coke samples were subjected to a series of reaction conditions, including the standard test and others with different high temperature and CO_2_ content. The standard test involves reacting 200 g of 23–25 mm coke particles at 1100 °C with 100% CO_2_ for 2 h according to the Chinese standard f GB/T 4000-2008, and the others were conducted with a gas mixture of CO_2_ and N_2_ under different temperatures and duration times. For all the tests, the flowing rate of total gases was 5 L/min. The reaction index of the weight loss percentage (RI) and tumbling strength after reaction (I_10_^600^) were tested according to the standard method.

The surface morphology was observed by scanning electron microscopy (SEM) (The Dutch FEI)) with an accelerating voltage of 20 kV. X-ray diffraction was performed on an ESCALAB 250Xi (Bruker, Germany) with Al Ka X-ray radiation as the X-ray source for excitation. The optical texture was examined on polished surfaces using a ZEISS Imager M1m optical microscope. The optical texture of coke was classified by isotropy and anisotropy.

## 3. Results and Discussion

### 3.1. RI/I_10_^600^ of Metallurgical Coke under Different Reaction Conditions

Table 2 shows the testing conditions and RI/I_10_^600^ results of metallurgical coke C1 after reaction with CO_2_. T1 is the standard test and the RI/I_10_^600^ under T1 is equal to CRI/CSR. In theory, the chemical reaction of coke and CO_2_ follows the relation of Equation (1), that is, 1 mol of C atoms needs 1 mol of CO_2_ to finish the reaction. With the assumption of coke being composed of 100% carbon, it may be approximately calculated that every 12 g coke by weight needs 22.4 L CO_2_ by volume. Table 2 also gives the total amount of CO_2_ fed, the reaction amount of CO_2_, and the reaction ratio of CO_2_. The reaction amount is calculated through 200 g coke multiplied by reactivity, divided by 12 g, and multiplied by 22.4 L. The reaction ratio of CO_2_ refers to the reaction amount divided by the amount fed. It can be seen that the RI values were 24.7%, 19.3%, and 12.9% respectively, from T1, T2, and T3 with the CO_2_ concentration reducing by half sequentially under the same duration time and temperature. Compared with T1, the RI values of T2 and T3 did not decrease by half sequentially and were far higher than the values calculated by half (12.35% and 6.175%). The reason is that the reaction ratio of CO_2_ increased from 15.4% to 24.0% and 32.2%. With the decrease in CO_2_ content, the reaction ratio of CO_2_ is increased because the chemical reaction is controlled by diffusion to some extent, and less CO_2_ was released without reaction under low composition. It can also be seen that by comparing T2 with T4 and T1 with T5 under the same CO_2_ content, with the increase in temperature and decrease in duration time, tiny differences existed for the RI values, but the reaction ratio of CO_2_ increased greatly. It can be noted that with the increasing temperature, the reaction rate accelerates and the reaction ratio of CO_2_ increases.
C + CO_2_ = 2CO(1)

Furthermore, the reaction tests of T1 and T5 contain the same CO_2_ concentration of 100% CO_2_ but different temperatures (1100 °C and 1300 °C) and duration times (2 h and 0.5 h). The time difference is 4× and the temperature is 200 °C, but the I_10_^600^ results are nearly the same of 68.6% and 68.1%. At the same time, according to the comparison of T2 and T4, the time difference is 2 and temperature is 100 °C, but the I_10_^600^ results are also nearly the same of 72.7% and 71.0%.

It can be concluded that temperature, gas composition, and reaction time all affect solution reaction and tumbling strength of coke. It may be presumed that the influence of increased temperature on tumbling strength for metallurgical coke C1 from a standard test of 1100 °C can be offset by reducing duration time with the relation of every 100 °C by half time.

### 3.2. RI/I_10_^600^ of Pilot Cokes from Single Coals under High Temperature and Standard Conditions

Because little change was found in the tumbling strength for metallurgical coke C1 under T1 of the standard test and T5 of high temperature, more samples of cokes from single coals were tested to further study the change relations. The results are shown in Table 3.

It can be seen that the RI/I_10_^600^ change results of cokes from single coals under high temperature of T5 are very different. The I_10_^600^ values may be smaller, larger, or similar compared to their CSR under the standard condition. For C2 and C3, the I_10_^600^ are smaller than their CSR, while for C5, C7, C8, and C9, the I_10_^600^ is larger, and for C4 and C6, the I_10_^600^ values are almost the same. The I_10_^600^ of C2 and C3 decreases moderately to 66.7% and 59.6% from 77.5% and 62.6%. The values of (I_10_^600^-CSR) are −10.8% and −3%, respectively. While the I_10_^600^ of C5, C7, C8, and C9 increases greatly to 65.9%, 63.8%, 63.7%, and 65.1% from 45.7%, 32.3%, 25.5%, and 14.7%, the values of (I_10_^600^-CSR) are 20.2%, 31.5%, 38.2%, and 50.4%, respectively. The I_10_^600^ of C4 and C6 changes slightly to 58.4% and 46.0% from 57.3% and 45.1% and the values of (I_10_^600^-CSR) are 1.1% and 0.9% respectively, which can be considered as no significant changes occurring.

As mentioned above, the influence of reaction temperature on tumbling strength of metallurgical C1 from a standard test can be offset by time with the relation of every 100 °C by half time; that is, the tumbling strength values of C1 under T1 and T5 were of a similar range. However, this seems to be inapplicable for different cokes from single coals here.

The comparison between I_10_^600^ (red data) under T5 with CSR (I_10_^600^ under T1, black data) is shown in Figure 1. It can be seen that the I_10_^600^ changes under high temperature are very different. Generally, the I_10_^600^ values for low CSR cokes (C7, C8, and C9) increase from conditions of T1 to T5. The lower the CSR, the higher the amount will increase. With the increase in CSR, the increasing amount of I_10_^600^ becomes smaller and smaller until it disappears. Some decreases may even occur for high CSR cokes (C2 and C3).

On the other hand, after reacting with CO_2_ at a low temperature of 1100 °C, cokes from different volatile coals have big differences in tumbling strength. While at a high temperature of 1300 °C, the differences shrink, and sometimes a reversion even appears (for red data, some left values of I_10_^600^ with low CSR are larger than those of the right ones).

Furthermore, the values of I_10_^600^ under T5 minus CSR under T1 (I_10_^600^-CSR) vs. CRI are shown in Figure 2. It appears that the (I_10_^600^-CSR) values increase with the increase in CRI, and a line relation may exist, indicating that for cokes with high CRI from high volatile coals, a strong resistance capacity of strength deterioration appears at high temperatures.

### 3.3. Morphology

Tumbling strength is highly related to the walls and pores of coke. As mentioned above, the strength of pilot cokes had great differences under T1 and various changes occurred from T1 to T5, especially for cokes of C2 and C9 (77.5% and 14.7% under T1 and 66.7% and 65.1% under T5). The cokes of C2 and C9 were chosen as examples to study the microstructure morphologies. Their SEM images are shown in Figure 3 and Figure 4. It can be clearly seen that the degradation of coke surface tends to be serious with the increase of reaction temperature. The raw of C9 has more developed pore structure with uneven pore distribution than that of C2 (Figure 3a and Figure 4a), which may be conducive for gas diffusion, providing advantageous conditions for the solution loss reaction of coke. After being reacted at 1100 °C, the edges and corners are obviously dissolved; meanwhile, the pore walls become thinner and a number of pores are connected, resulting in some larger pores (Figure 3b or Figure 4b). This phenomenon for C9 is more obvious than that of C2, which may explain the very high CRI and low CSR of C9 (66.2%/14.7%). After reacted at 1300 °C, in comparison, all the edges and corners almost disappeared. Simultaneously, the walls of the pores are thinner and the extension of the pore connections is deeper than that of raw coke and coke reacted at 1100 °C (Figure 3c or Figure 4c).

### 3.4. Carbon Structure

Coke is created by the carbonization of coking coals and the average carbon crystal structure can be represented by a large number of small hexagonal crystallites in a turbostratic structure, along with small amounts of mineral matter as ash impurities [16]. The increase in structural order can be reflected in the increased narrowing of (002) peak by XRD characterization.

The XRD spectrums of C2 and C9 before reaction (raw samples) and after reaction at T1 and T5 are shown in Figure 5, respectively. For the (002) diffraction peaks of the two cokes, the intensity increases gradually and the shape sharpens with the increase in reaction temperature.

The carbon crystal structure parameters are listed in Table 4. The interplanar spacing (d(002)), crystallite size (La), and stack height of carbon crystallite (Lc) of coke are calculated using the classical Scherrer equation where *n* is the number of graphitic planes in the stacking crystallite [14]. The value of (n_T5_ − n_Raw_)/nRaw is the *n* growth of coke under T5 compared to the original coke. It can be seen that the Lc values increase and the d(002) values decrease with the increase of temperature for most cokes. The *n* values increase as the temperature increases, which is especially apparent under T5 with the growth range from 49.23% to 95.18%.

### 3.5. Optical Texture

Coke is a porous, fissured material which consists of pores, microfissures, and a solid carbon matrix with organic and inorganic inclusions. Recent research has shown that the properties of metallurgical coke depend on the relative proportion of isotropic carbon and inert, the size and shape of the anisotropic carbon units, the interface among textural components, porosity, and ash chemistry [16,17]. Many scientists also studied the consequences of anisotropy and the isotropic effect on coke quality and other technological parameters. They found that the reactivity of coke has a tendency to increase with the increase in isotropy. This indicates that a coke solution with carbon dioxide over a temperature of 800 °C shows selective attack, with the isotropic texture reacting more readily than the anisotropic texture [18,19].

According to vitrinite reflectance distribution, the optical texture of coke is mainly divided into isotropy (represented as ΣISO) and anisotropy (represented as ΣOTI). The optical texture components of raw cokes are illustrated in Figure 6. It can be seen that C5, C7, C8, and C9 contain more ΣISO than other cokes, and their ΣOTI is relatively less. On the contrary, C2, C3, C4, and C6 contain more ΣOTI and less ΣISO.

The ΣISO components of raw coke after reaction under T1 and T5 are shown in Figure 7. It can be seen that the ΣISO composition of all the cokes after reaction under T1 is less than that of raw cokes and cokes reacted under T5, indicating that the isotropy of coke is easier to react with CO_2_ than the anisotropy under T1 at 1100 °C [20].

The reaction between carbon and CO_2_ mainly depends on activated carbons for adsorption and reaction, which are located at the edges and corners of coke. Isotropic carbon layers with random stacking have more micropores and activated carbon atoms, which makes it easier adsorb CO_2_ for reaction. The carbon layers of anisotropic structures with orderly stacking have fewer micropores and activated carbons and do not easily adsorb CO_2_ for reaction. It is for this reason that the isotropy of coke is easier to react with CO_2_ than the anisotropy at 1100 °C [21,22].

The ΣISO ratio of cokes under T5 is larger than that of T1, indicating that the anisotropy reacts easier with CO_2_ than the isotropy at 1300 °C when compared with 1100 °C. It can be considered that at high temperature, the carbon atoms in anisotropic structures of ordered stacking can easily react with Na, K in coke to produce intercalation compounds, which would increase the distance and surface area of the reaction between the carbon layer, resulting in an easier reaction with CO_2_ [23].

The I_10_^600^ (under T1 and T5) and ΣISO components (raw coke) for all the cokes are displayed in Figure 8. It can be seen that the I_10_^600^ under T5 of C5, C7, C8, and C9 with high ΣISO values are far higher than their CSR. The I_10_^600^ under T5 of C4 and C6 with medium ΣISO are similar with their CSR, while the I_10_^600^ of C2 and C3 with low ΣISO values are lower than their CSR. These results illustrate that the ΣISO components of cokes have a serious effect on the I_10_^600^ at different temperatures. With the increase of temperature from 1100 °C to 1300 °C, the ΣISO components of cokes is first easier to react with CO_2_ at 1100 °C, and then becomes harder at 1300 °C, as compared with ΣOTI. It can be considered that at a high temperature, with the reacting of carbon atoms in anisotropic structures of ordered stacking with Na, K, some cracks in the carbon layer are created. The carbon atoms in unordered stacked isotropy are not so easy to react with Na, K to produce intercalation compounds. Thus, the tumbling strength of isotropy is stronger than that of anisotropy at high reaction temperatures [23].

Figure 9 shows the *n* and ΣOTI (raw coke) values for all the cokes. It can be seen that the cokes of C2, C3, C4, and C6 containing more raw ΣOTI components have a higher *n* growth ratio under T1 and T5 as compared with C5, C7, C8, and C9 containing less ΣOTI components. It can be considered that the cokes which contain more anisotropy are easier to graphitize than other cokes with less anisotropy under high temperature [24].

## 4. Conclusions

The evaluating accuracy of CRI/CSR for coke in BFs has been disputed for many years. Many researchers have revised the CRI and CSR test conditions, including changing the pure CO_2_ to a mixture, raising the reaction temperature to high temperatures, controlling a constant weight loss of coke, etc. Most of the revised methods are either too complicated to be carried out or are time consuming. In this paper, a simple method of a 1300 °C and 0.5 h high temperature test is proposed to replace complicated simulation methods. Almost the same differences were found for evaluating coke quality between simulated and standard conditions. Temperature greatly affects the solution reaction and tumbling strength after coke reaction; especially at high temperatures up to 1300 °C, the reactivity of coke increases greatly. Under a high temperature of 1300 °C and a duration time of 0.5 h, the changes of RI/I_10_^600^ for cokes from different coals are very different compared with the standard test, and greatly related to the optical texture of cokes. Cokes with more isotropy increase in strength at high temperatures, and decrease with less isotropy. Similar to the methods proposed by other researchers, there is still a need for further experiments of more examples of different cokes. Furthermore, the accuracy of the method should be examined by the actual experience of production in the future.

## Figures and Tables

**Figure 1 materials-14-05767-f001:**
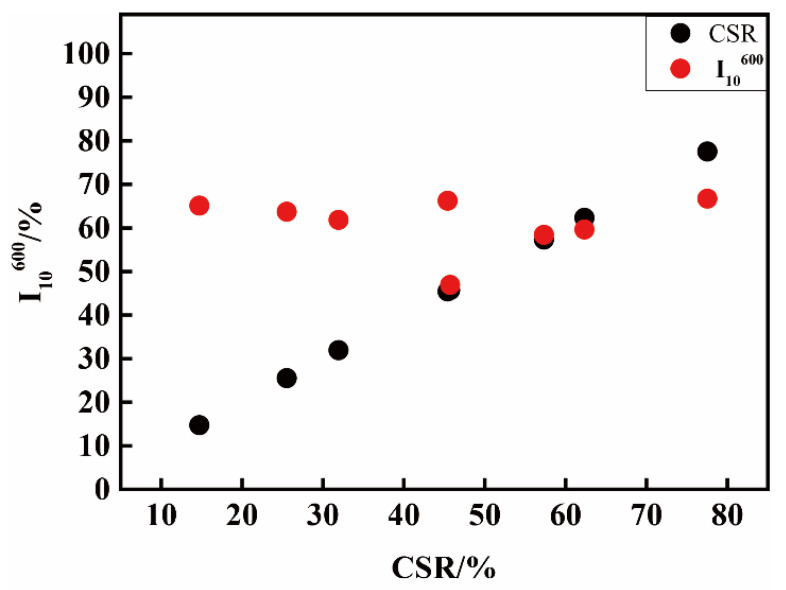
The comparison of I_10_^600^ under T5 with CSR under T1 of cokes from single coals.

**Figure 2 materials-14-05767-f002:**
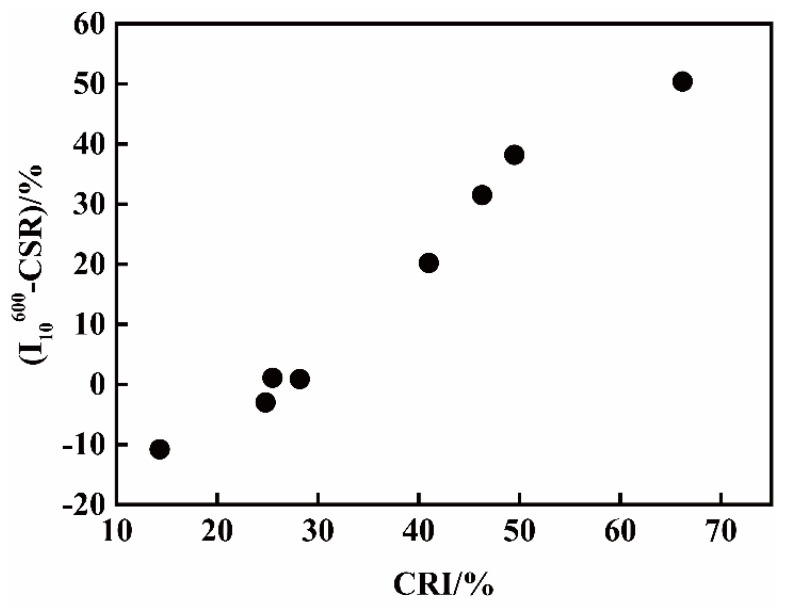
The (I_10_^600^-CSR) vs. CRI of cokes.

**Figure 3 materials-14-05767-f003:**
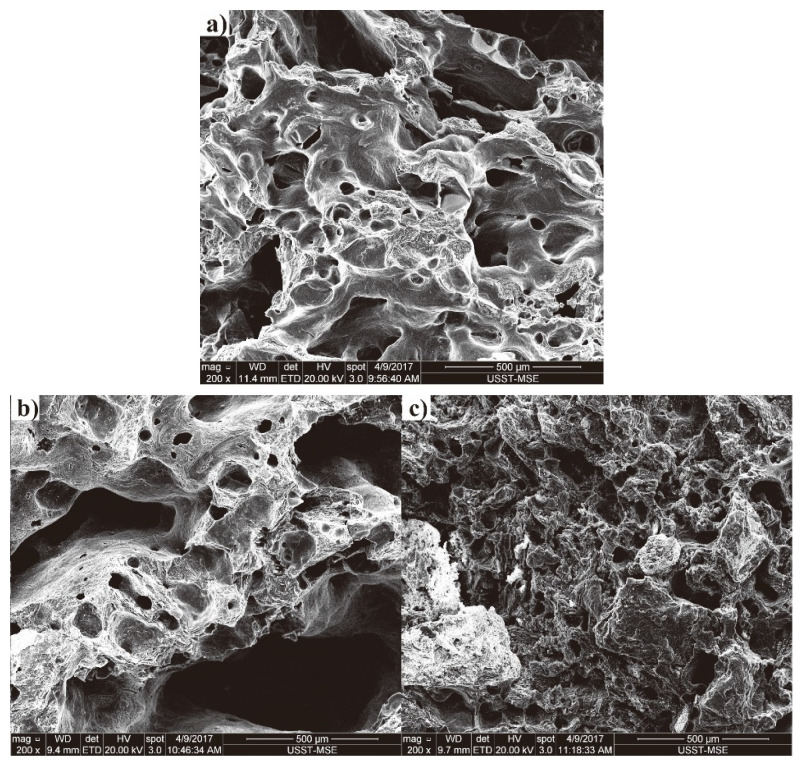
The SEM images of C2: (**a**) raw, (**b**) larger pores; (**c**) after reaction under T1 and T5.

**Figure 4 materials-14-05767-f004:**
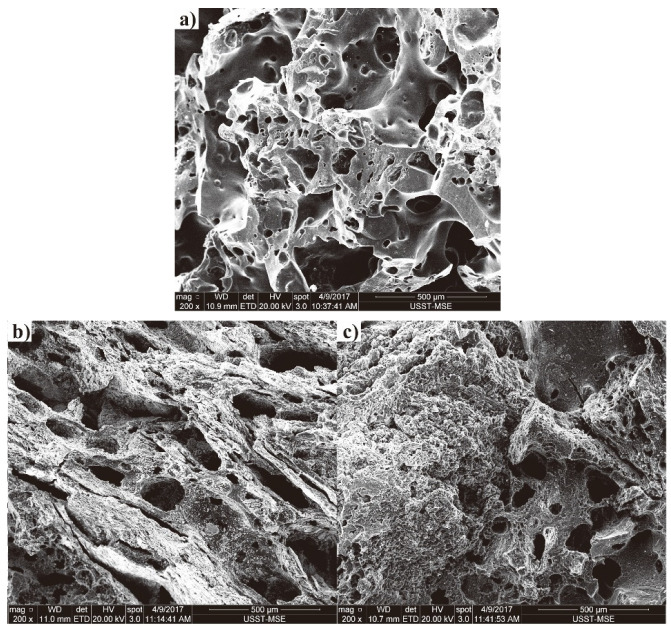
The SEM images of C9: (**a**) raw, (**b**) and (**c**) after reaction under T1 and T5.

**Figure 5 materials-14-05767-f005:**
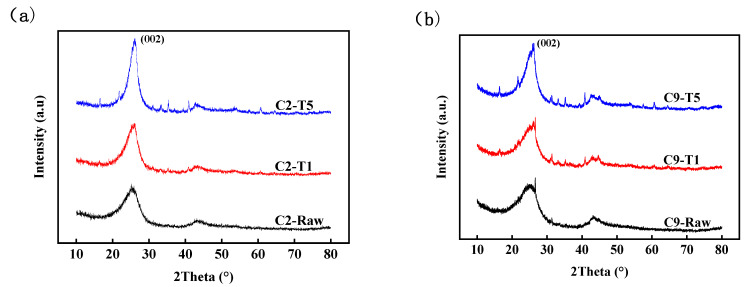
XRD diffraction spectrums for: (**a**) C2 and (**b**) C9.

**Figure 6 materials-14-05767-f006:**
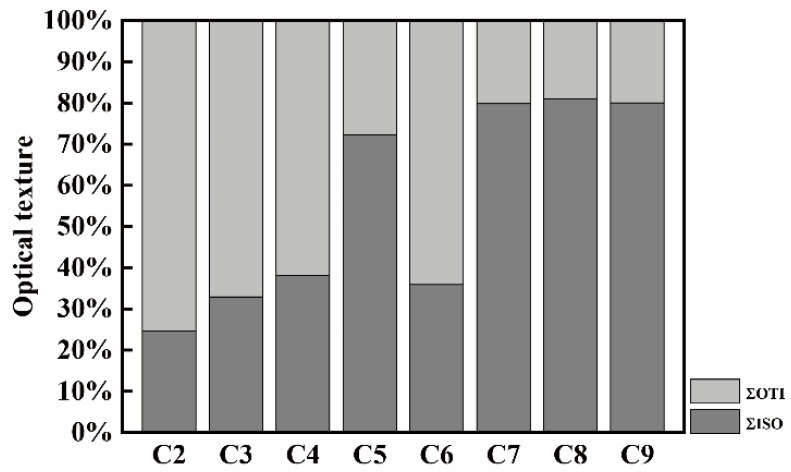
The optical texture of raw cokes.

**Figure 7 materials-14-05767-f007:**
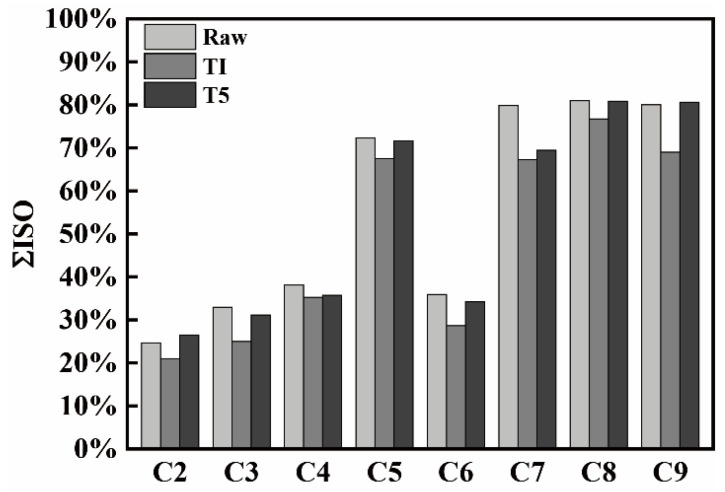
The ΣISO of cokes under different conditions.

**Figure 8 materials-14-05767-f008:**
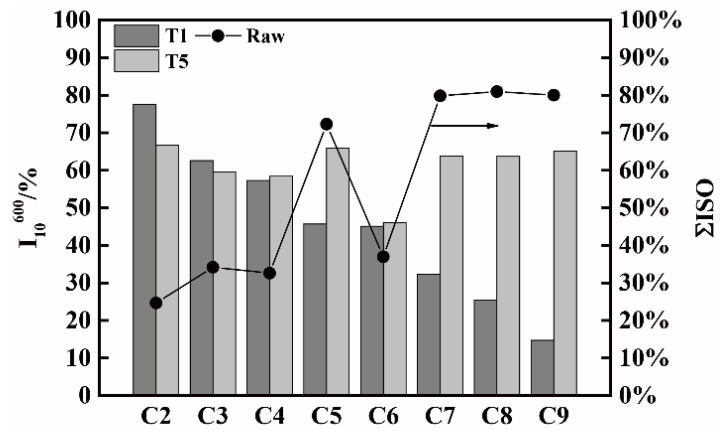
The I_10_^600^ (under T1 and T5) and ΣISO (raw coke) values for all the cokes.

**Figure 9 materials-14-05767-f009:**
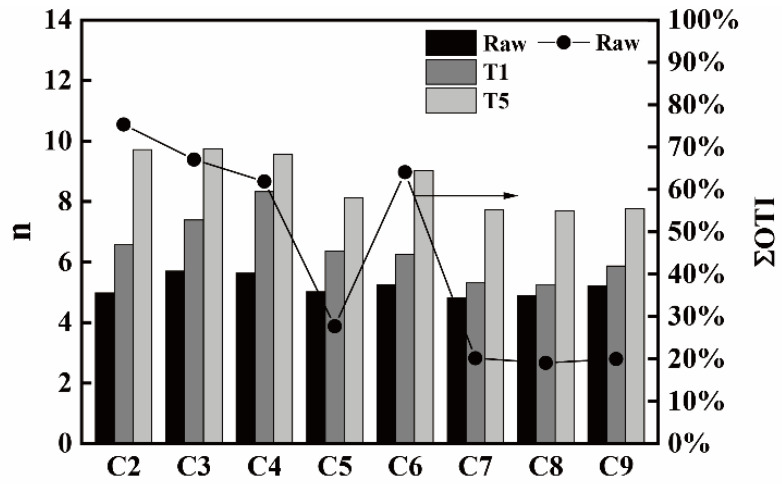
The *n* and ΣOTI (raw coke) values for all the cokes.

**Table 1 materials-14-05767-t001:** Proximate analyses of the coals.

Items	C1	C2	C3	C4	C5	C6	C7	C8	C9
Moisture, %	8.68	9.78	9.90	9.20	11.05	9.60	7.30	9.72	8.20
Fixed carbon, %	64.63	69.94	68.96	61.42	65.14	69.72	57.20	57.01	57.55
Ash, %	8.96	9.98	10.03	9.13	9.94	9.45	7.93	8.06	7.58
Volatile, %	26.41	20.08	21.01	29.45	24.93	20.81	34.86	34.93	34.87
G	84.00	90.75	86.00	90.00	90.50	75.43	80.60	79.80	78.70
Y, mm	14.00	14.50	11.00	23.00	23.75	10.14	11.40	13.00	13.33
A + B, %	43.50	81.25	58.00	176.00	113.50	11.71	29.80	39.00	122.50

**Table 2 materials-14-05767-t002:** Test conditions and RI/I_10_^600^ results of metallurgical coke C1.

Test	Test Conditions				C1			
	Temperature (°C)	Time (h)	CO_2_ (%)	Total Amount of CO_2_ Fed (L)	RI (%)	I_10_^600^ (%)	Reaction Amount of CO_2_ (L)	Reaction Ratio of CO_2_ (%)
T1 ^a^	1100	2	100%	600	24.7	68.6	92.2	15.4
T2	1100	2	50%	300	19.3	72.7	72.0	24.0
T3	1100	2	25%	150	12.9	78.4	48.2	32.2
T4	1200	1	50%	150	18.1	71.0	67.6	45.0
T5	1300	0.5	100%	150	23.3	68.1	87.0	58.0

^a^ T1 is the standard test.

**Table 3 materials-14-05767-t003:** The results of RI/I_10_^600^ (%/%) for pilot cokes from single coals under high temperature and standard conditions.

Test	C2	C3	C4	C5	C6	C7	C8	C9
T1 ^a^ (CRI/CSR)	14.3/77.5	24.8/62.6	25.5/57.3	41.0/45.7	28.2/45.1	46.3/32.3	49.5/25.5	66.2/14.7
T5 (RI/I_10_^600^)	19.3/66.7	26.0/59.6	28.6/58.4	29.7/65.9	28.4/46.0	31.4/63.8	34.4/63.7	34.3/65.1
(I_10_^600^-CSR)	−10.8%	−3%	1.1%	20.2%	0.9%	31.5%	38.2%	50.4%

^a^ The RI/I_10_^600^ of T1 is CRI/CSR.

**Table 4 materials-14-05767-t004:** The carbon crystal structure parameters of cokes.

Coke.		β(002) (°)	d(002) (nm)	Lc (nm)	La (nm)	*n* (Lc/d(002))	(n_T5_ − n_Raw_)/n_Raw_ (%)
C2	Raw	4.694	0.3534	1.756	5.644	4.98	95.18
	T1	3.658	0.3438	2.256	6.054	6.57	
	T5	2.482	0.3423	3.326	7.610	9.72	
C3	Raw	4.282	0.3387	1.929	8.611	5.70	70.88
	T1	3.316	0.3373	2.491	7.886	7.39	
	T5	2.519	0.3371	3.280	9.401	9.74	
C4	Raw	4.382	0.3354	1.886	5.959	5.63	78.54
	T1	2.953	0.3362	2.799	7.994	8.33	
	T5	2.508	0.3439	3.290	8.228	9.57	
C5	Raw	4.661	0.3521	1.769	6.041	5.02	61.75
	T1	3.798	0.3423	2.174	6.875	6.36	
	T5	2.958	0.3437	2.790	8.244	8.12	
C6	Raw	4.482	0.3497	1.840	5.954	5.25	71.81
	T1	3.792	0.3469	2.175	9.086	6.26	
	T5	2.663	0.3441	3.099	9.402	9.02	
C7	Raw	4.892	0.3496	1.686	8.146	4.82	60.63
	T1	3.440	0.3424	2.400	5.265	5.31	
	T5	2.783	0.3396	2.628	5.185	7.74	
C8	Raw	5.052	0.3354	1.636	5.545	4.88	57.79
	T1	4.595	0.3407	1.797	4.898	5.26	
	T5	2.479	0.3396	2.620	5.000	7.70	
C9	Raw	4.730	0.3362	1.747	3.698	5.20	49.23
	T1	4.002	0.3507	2.060	4.939	5.87	
	T5	3.133	0.3395	2.636	5.268	7.76	

## Data Availability

The data presented in this article are available on request from the corresponding author.

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
