# Peer review of "A Simple Method of Evaluating the Thermal Properties of Metallurgical Cokes under High Temperature"

_materials, 2021, doi:10.3390/ma14195767_

Round 1
Reviewer 1 Report
Materials
A simple method of evaluating the thermal properties of metallurgical cokes under high temperature
Summary: The submission is a systematic experimental study for evaluating coke properties by standard tests and newly proposed replacements. The major conclusion is that higher temperatures and reduced CO2 concentrations could be used to determine reactivity and strength to save time. With major revisions and clarifications the submission could be published in Materials.
Intro
1st paragraph English could be improved for this sentence “Coke with appropriate reactivity and high strength is very important which affects BF greatly in stable running.”
2nd paragraph – the background on shortcomings of the standard test for representing BF conditions is a good start. Please add more discussion on the arguments made in your references [5-14] on why CRI/CSR could be improved.
Methods
This reviewer does not understand how “amount of CO2 needed” is determined. Is this simply tied to the mass of the sample ((mass/12.01*1/1)?
Please elaborate on the benefit to using N2 to dilute the CO2 stream. I can see using diluting CO2 can help make the experimental conditions more comparable but the five tests T1-T5 in table 2 have some many variations that temperature, dilution and time all change randomly and trends are harder to observe with multiple parameters being adjusted.
Please improve the resolution quality for the figure images. The current version has blurry figures.
XRD – please provide references, assumptions and input parameters for Scherrer, La and LC parameter analysis.
Conclusions:
First paragraph: Do mean 1300C, 0.5h and 100% CO2 is a simple method to replace NSC?
Please provide more background and guidance on how your proposed tests could be accepted, widely implemented and finally replace the standard NSC test.
Author Response
Reviewer 1
A simple method of evaluating the thermal properties of metallurgical cokes under high temperature
Summary: The submission is a systematic experimental study for evaluating coke properties by standard tests and newly proposed replacements. The major conclusion is that higher temperatures and reduced CO2 concentrations could be used to determine reactivity and strength to save time. With major revisions and clarifications the submission could be published in Materials.
Intro
Question 1: 1st paragraph English could be improved for this sentence “Coke with appropriate reactivity and high strength is very important which affects BF greatly in stable running.”
Our response: Thank you for the advice. It has been replaced by the sentence “It is very important to use the cokes with appropriate reactivity and high strength for stable running of BF.”
Question 2: 2nd paragraph – the background on shortcomings of the standard test for representing BF conditions is a good start. Please add more discussion on the arguments made in your references [5-14] on why CRI/CSR could be improved.
Our response: Thank you for the advice. We have added some sentences as follows:
Many researchers have revised the CRI and CSR test conditions, including changing the pure CO2 to mixture, raising the reaction temperature to high temperature, controlling a constant weight loss of coke and so on. Most of them found a poor correlation between the results of simulated conditions and standard conditions, or even a mistake for the estimation of coke quality by CRI/CSR. They proposed that the post reaction strength test should be modified in accordance with the individual BF operation, especially at high temperature.
In our former publication of reference [15], we have also reported a simulated BF conditions as Table 1. And we also found that compared with standard CRI/CSR test, the changes of reactivity and tumbler strength of cokes made from different coals are very different under simulated BF conditions. The I10600 after reaction of low CSR coke under simulated BF conditions improves significantly. In the followed study we investigated more samples in our laboratory and found the same results, which further confirms that the standard NSC method does have some limitations in simulating coke degradation behavior in BF and evaluating the thermal properties of coke.
However, the reaction temperature is up to 1350 ℃ and the whole experiment time is about 334 min. It is more complicated compared with standard method. In our opinion, the difference of temperature is the most important factor that affects the evaluation accuracy of coke. We try to find a simple method for high temperature test in this manuscript.
Table 1 BF heating rate and gas composition specifications
|
Temperature (℃) |
Heating rate (℃/min) |
Total time (min) |
Gas composition and volume |
|
|
CO2 (L/min) |
N2 (L/min) |
|||
|
20-800 |
9 |
85 |
0 |
2 |
|
800-1200 |
2 |
200 |
0.5 |
3.5 |
|
1150 |
hold |
19 |
3 |
1 |
|
1200-1300 |
5 |
20 |
0.5 |
3.5 |
|
1300-1350 |
5 |
10 |
3 |
1 |
Question 3: Methods
This reviewer does not understand how “amount of CO2 needed” is determined. Is this simply tied to the mass of the sample ((mass/12.01*1/1)?
Our response: Thank you for having pointed out the contents of hard understanding.
In the first manuscript, the “amount of CO2 needed” means the reaction amount of CO2 in Table 2. We have revised it as “reaction amount of CO2”. With the assumption of coke being composed by 100% carbon, it is approximately calculated that every 12g coke by weight needs 22.4L CO2 by volume for reaction. The reaction amount of CO2 is calculated through 200g coke multiplied by reactivity, divided by 12g and multiplied by 22.4L. The Reaction ratio of CO2 refers to the reaction amount divided by the amount fed.
By approximately calculating the reaction ratio of CO2, we could see how many percent of CO2 reacted under different conditions.
We have added the illustrations of the above sentences in the new manuscript.
Question 4: Please elaborate on the benefit to using N2 to dilute the CO2 stream. I can see using diluting CO2 can help make the experimental conditions more comparable but the five tests T1-T5 in table 2 have some many variations that temperature, dilution and time all change randomly and trends are harder to observe with multiple parameters being adjusted.
Our response: Thank you for the questions.
In the actual blast furnace, the gas is a mixture and mainly contains N2, CO, H2, H2O and CO2. The reaction condition of standard test for CRI/CSR is different from the actual experiences of coke in BF at temperature, composition, and reaction time. Indeed, the difference of temperature is the most important factor that affects the evaluation accuracy of coke. Here we want to study the key factor of temperature instead of the composition of gas such as the introduce of other reaction gas of H2 or H2O. With the increase of temperature, the reaction accelerates quickly. The feed of CO2 or the reaction time must be reduced to keep adequate coke unreacted for evaluation. We use the methods of diluting CO2 and reducing the reaction time for high temperature reaction to decreasing the reaction amount of coke. N2 is one of the inert gas in the blast furnace which is also cheap, safe and convenient in laboratory, which is the reason we used N2 as the diluting gas for CO2. For high temperature experiment, although the diluting of CO2 tends to decrease the reaction amount of coke, it doesn’t benefit the reducing of reaction time. We recommend the use of pure CO2 for under a short time for high temperature research.
Question 5: Please improve the resolution quality for the figure images. The current version has blurry figures.
Our response: Thank you for the suggestions and we have revised it for improving the resolution quality.
Question 6: XRD – please provide references, assumptions and input parameters for Scherrer, La and LC parameter analysis.
Our response: The crystallite size (La) and stack height of carbon crystallite (Lc) of coke are calculated using classical Scherrer Equation, n is the number of graphitic planes in the stacking crystallite. The reference [14] provided the details of the information and has been cited in the new manuscript.
Question 7: Conclusions:
First paragraph: Do mean 1300C, 0.5h and 100% CO2 is a simple method to replace NSC?
Please provide more background and guidance on how your proposed tests could be accepted, widely implemented and finally replace the standard NSC test.
Our response: Thank you for the questions.
The reaction condition of standard test for CRI/CSR is different from the actual experiences of coke in blast furnace at temperature, composition, and reaction time. Many researchers have revised the CRI and CSR test conditions, including changing the pure CO2 to mixture, raising the reaction temperature to high temperature, controlling a constant weight loss of coke and so on. Most of them found a poor correlation between the results of simulated conditions and standard conditions, or even a mistake for the estimation of coke quality by CRI/CSR.
We have also reported a simulated BF conditions and found that compared with standard CRI/CSR test, the changes of reactivity and tumbler strength of cokes made from different coals are very different under simulated BF conditions. The reaction temperature is up to 1350 ℃ and the whole experiment time is about 334 min. It is more complicated compared with standard method. In our opinion, the difference of temperature is the most important factor that affects the evaluation accuracy of coke. We try to find a simple method for high temperature test in this manuscript.
As for the experimental coke samples in the manuscript, we obtained almost the same results compared with the complicated simulated conditions. We proposed here the method of 1300℃ and 0.5h for high temperature as a simple method to replace complicated ones.
As for the question of whether it can be used to replace the method of NSC, in my opinion, like the methods proposed by other researchers, there still need lots of experiments for more examples of different cokes. The accuracy of the method should be examined by the actual experience of production. And also, more scholars who do the research of this fields are welcome to propose their own findings and points.
The key opinions here have been added in the conclusions of the new manuscript.

Reviewer 2 Report
- It could be appropiate to explain reaction amount of CO2 and reaction ratio of CO2.
- Reaction ratio of CO2 - it should be stated how this is determined.
- Is it right to mention CO2 composition? Is it composition in that sense?
- Table 3 is not easy readable (for me).
- Fig. 5 - there is not evident labeling of single coals.
- In the text "black data" and "red data" is explained late. It should be done one paragraph earlier.
- Fig. 2 - there is not evident labeling of single coals.
- Which factors (listed in the table 1) may affect isotropy?
Author Response
Reviewer 2
Question 1: It could be appropriate to explain reaction amount of CO2 and reaction ratio of CO2.
Our response: Thank you for the advice. The reaction amount is calculated though 200g coke multiplied by reactivity, divided by 12g and multiplied by 22.4 L. The Reaction ratio of CO2 refers to the reaction amount divided by the amount fed. We have added the explanations in the new manuscript.
Question 2: Reaction ratio of CO2 - it should be stated how this is determined.
Our response: Thank you for the advice. The Reaction ratio of CO2 refers to the reaction amount divided by the amount fed. We have added the explanations in the new manuscript.
Question 3: Is it right to mention CO2 composition? Is it composition in that sense?
Our response: Thank you for the question. The CO2 composition should be replaced by CO2 content. We have revised them in the new manuscript.
Question 4: Table 3 is not easy readable (for me).
Our response: Thank you for the question. We have added the (CRI/CSR) for T1 and (RI/I10600) for T5 to help readers know about the data. We have also added the unit (%) of the data in the title of the Table.
Question 5: Fig. 5 - there is not evident labeling of single coals.
Our response: Fig. 5 shows the XRD spectrums of C2 and C9 as examples of the cokes before reaction (raw samples), after reaction at T1 and T5, respectively. We have labeled every samples as sample-T5, -T1, and -Raw, respectively.
Question 6: In the text "black data" and "red data" is explained late. It should be done one paragraph earlier.
Our response: Thank you for the suggestion. We have revised it as advised.
Question 7: Fig. 2 - there is not evident labeling of single coals.
Our response: In Fig.2, the values of I10600 under T5 minus CSR under T1 (I10600-CSR) vs CRI are shown. We want to show readers that the (I10600-CSR) values increase with the increase of CRI and a line relation may exist, which indicates that the cokes with high CRI from high volatile coals have a strong resistance capacity of strength deterioration at high temperature appears. We want the readers to find the relations of different cokes and forget the name of every specific samples. I recommend Fig.2 to be kept as current appearance.
Question 8: Which factors (listed in the table 1) may affect isotropy?
Our response: According to vitrinite reflectance distribution, the optical texture of coke is mainly divided into isotropy (represented as ΣISO.) and anisotropy (represented as ΣOTI). The isotropy structure of coke is mainly classified by the vitrinite reflectance. According to the factors listed in Table 1, the isotropy of coke is mainly affected by the volatile of the coal. Usually the coal with a high volatile will have a high Coke ΣISO component after being coked.

Round 2
Reviewer 1 Report
thanks